# Task-Driven Convolutional Recurrent Models of the Visual System

**Aran Nayebi[1,*], Daniel Bear[2,*], Jonas Kubilius[5,7,*], Kohitij Kar[5], Surya Ganguli[4,8], David Sussillo[8], James J. DiCarlo[5,6], and Daniel L. K. Yamins[2,3,9]**

[1]Neurosciences PhD Program, Stanford University, Stanford, CA 94305
[2]Department of Psychology, Stanford University, Stanford, CA 94305
[3]Department of Computer Science, Stanford University, Stanford, CA 94305
[4]Department of Applied Physics, Stanford University, Stanford, CA 94305
[5]McGovern Institute for Brain Research, MIT, Cambridge, MA 02139
[6]Department of Brain and Cognitive Sciences, MIT, Cambridge, MA 02139
[7]Brain and Cognition, KU Leuven, Leuven, Belgium
[8]Google Brain, Google, Inc., Mountain View, CA 94043
[9]Wu Tsai Neurosciences Institute, Stanford, CA 94305
[*]Equal contribution. {anayebi,dbear}@stanford.edu; qbilius@mit.edu

## Abstract

Feed-forward convolutional neural networks (CNNs) are currently state-of-the-art for object classification tasks such as ImageNet. Further, they are quantitatively accurate models of temporally-averaged responses of neurons in the primate brain's visual system. However, biological visual systems have two ubiquitous architectural features not shared with typical CNNs: local recurrence within cortical areas, and long-range feedback from downstream areas to upstream areas. Here we explored the role of recurrence in improving classification performance. We found that standard forms of recurrence (vanilla RNNs and LSTMs) do not perform well within deep CNNs on the ImageNet task. In contrast, novel cells that incorporated two structural features, bypassing and gating, were able to boost task accuracy substantially. We extended these design principles in an automated search over thousands of model architectures, which identified novel local recurrent cells and long-range feedback connections useful for object recognition. Moreover, these task-optimized ConvRNNs matched the dynamics of neural activity in the primate visual system better than feedforward networks, suggesting a role for the brain's recurrent connections in performing difficult visual behaviors.

## 1   Introduction

The visual system of the brain must discover meaningful patterns in a complex physical world [James, 1890]. Animals are more likely to survive and reproduce if they reliably notice food, signs of danger, and memorable social partners. However, the visual appearance of these objects varies widely in position, pose, contrast, background, foreground, and many other factors from one occasion to the next: it is not easy to identify them from low-level image properties [Pinto et al., 2008]. In primates, the visual system solves this problem by transforming the original "pixel values" of an image into a new internal representation, in which high-level properties are more explicit [DiCarlo et al., 2012]. This can be modeled as an algorithm that encodes each image as a set of nonlinear features. For an animal, a good encoding algorithm is one that allows behaviorally-relevant information to be decoded simply from these features, such as by linear classification [Hung et al., 2005, Majaj et al., 2015]. Understanding this encoding is key to explaining animals' visual behavior [Rajalingham et al., 2018].

Task-optimized, deep convolutional neural networks (CNNs) have emerged as quantitatively accurate models of encoding in primate visual cortex [Yamins et al., 2014, Khaligh-Razavi and Kriegeskorte, 2014, Güçlü and van Gerven, 2015]. This is due to (1) their cortically-inspired architecture, a cascade of spatially-tiled linear and nonlinear operations; and (2) their being optimized to perform the same behaviors that animals must perform to survive, such as object recognition [Yamins and DiCarlo, 2016]. CNNs trained to recognize objects in the ImageNet dataset predict the time-averaged neural responses of cortical neurons better than any other model class. Model units from lower, middle, and upper convolutional layers, respectively, provide the best-known linear predictions of time-averaged visual responses in neurons of early (area V1 [Khaligh-Razavi and Kriegeskorte, 2014, Cadena et al., 2017]), intermediate (area V4 [Yamins et al., 2014]), and higher visual cortical areas (inferotemporal cortex, IT, [Khaligh-Razavi and Kriegeskorte, 2014, Yamins et al., 2014]) These results are especially striking given that the parameters of task-optimized CNNs are entirely determined by the high-level categorization goal and never directly fit to neural data.

The choice to train on a high-variation, challenging, real-world task, such as object categorization on the ImageNet dataset, is crucial. So far, training CNNs on easier, lower-variation object recognition datasets [Hong et al., 2016] or unsupervised tasks (e.g. image autoencoding [Khaligh-Razavi and Kriegeskorte, 2014]) has not produced accurate models of per-image neural responses, especially in higher visual cortex. This may indicate that capturing the wide variety of visual stimuli experienced by primates is critical for building accurate models of their visual systems.

While these results are promising, complete models of the visual system must explain not only time-averaged responses, but also the complex temporal dynamics of neural activity. Such dynamics, even when evoked by static stimuli, are likely functionally relevant [Gilbert and Wu, 2013, Kar et al., 2018, Issa et al., 2018]. However, it is not obvious how to extend the architecture and task-optimization of CNNs to the case where responses change over time. Nontrivial dynamics result from biological features not present in strictly feedforward CNNs, including synapses that facilitate or depress, dense local recurrent connections within each cortical region, and long-range connections between different regions, such as feedback from higher to lower visual cortex [Gilbert and Wu, 2013]. Furthermore, the behavioral roles of recurrence and dynamics in the visual system are not well understood. Several conjectures are that recurrence "fills in" missing data, [Spoerer et al., 2017, Michaelis et al., 2018, Rajaei et al., 2018, Linsley et al., 2018] such as object parts occluded by other objects; that it "sharpens" representations by top-down attentional feature refinement, allowing for easier decoding of certain stimulus properties or performance of certain tasks [Gilbert and Wu, 2013, Lindsay, 2015, McIntosh et al., 2017, Li et al., 2018, Kar et al., 2018]; that it allows the brain to "predict" future stimuli (such as the frames of a movie) [Rao and Ballard, 1999, Lotter et al., 2017, Issa et al., 2018]; or that recurrence "extends" a feedforward computation, reflecting the fact that an unrolled recurrent network is equivalent to a deeper feedforward network that conserves on neurons (and learnable parameters) by repeating transformations several times [Liao and Poggio, 2016, Zamir et al., 2017, Leroux et al., 2018, Rajaei et al., 2018]. Formal computational models are needed to test these hypotheses: if optimizing a model for a certain task leads to accurate predictions of neural dynamics, then that task may be a primary reason those dynamics occur in the brain.

We therefore attempted to extend the method of goal-driven modeling from solving tasks with feedforward CNNs [Yamins and DiCarlo, 2016] or RNNs [Mante et al., 2013] to explain dynamics in the primate visual system, resulting in convolutional recurrent neural networks (ConvRNNs). In particular, we hypothesized that adding recurrence and feedback to CNNs would help these models perform an ethologically-relevant task and that such an augmented network would predict the fine-timescale trajectories of neural responses in the visual pathway.

Although CNNs augmented with recurrence have been used to solve some simple occlusion and future-prediction tasks [Spoerer et al., 2017, Lotter et al., 2017], these models have not been shown to generalize to harder tasks already performed by feedforward CNNs — such as recognizing objects in the ImageNet dataset [Deng et al., 2009, Krizhevsky et al., 2012] — nor have they explained neural responses as well as ImageNet-optimized CNNs. For this reason, we focused here on building ConvRNNs at a scale capable of performing the ImageNet object recognition task. Because of its natural variety, the ImageNet dataset contains many images with properties hypothesized to recruit recurrent processing (e.g. heavy occlusion, the presence of multiple foreground objects, etc.). Moreover, some of the most effective recent solutions to ImageNet (e.g. the ResNet models [He et al., 2016]) repeat the same structural motif across many layers, which suggests that they might perform computations similar to unrolling shallower recurrent networks over time [Liao and Poggio,

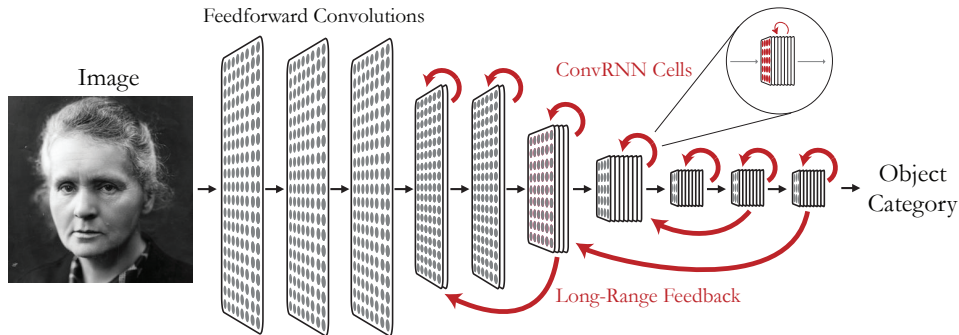

Figure 1: **Schematic of model architecture.** Convolutional recurrent networks (ConvRNNs) have a combination of local recurrent cells and long-range feedback connections added on top of a CNN backbone. In our implementation, propagation along each black or red arrow takes one time step (10 ms) to mimic conduction delays between cortical layers.

2016]. Although other work has used the output of CNNs as input to RNNs to solve visual tasks such as object segmentation [McIntosh et al., 2017] or action recognition [Shi et al., 2018], here we integrated and optimized recurrent structures within the CNN itself.

We found that standard recurrent motifs (e.g. vanilla RNNs, LSTMs [Elman, 1990, Hochreiter and Schmidhuber, 1997]) do not increase ImageNet performance above parameter-matched feedforward baselines. However, we designed novel local cell architectures, embodying structural properties useful for integration into CNNs, that do. To identify even better structures within the large space of model architectures, we then performed an automated search over thousands of models that varied in their locally recurrent and long-range feedback connections. Strikingly, this procedure discovered new patterns of recurrence not present in conventional RNNs. For instance, the most successful models exclusively used depth-separable convolutions for their local recurrent connections. In addition, a small subset of long-range feedback connections boosted task performance even as the majority had a negative effect. Overall this search resulted in recurrent models that matched the performance of a much deeper feedforward architecture (ResNet-34) while using only $\sim 75\%$ as many parameters. Finally, comparing recurrent model features to neural responses in the primate visual system, we found that ImageNet-optimized ConvRNNs provide a quantitatively accurate model of neural dynamics at a 10 millisecond resolution across intermediate and higher visual cortex. These results offer a model of how recurrent motifs might be adapted for performing object recognition.

## 2   Methods

**Computational Architectures.**   To explore the architectural space of ConvRNNs and compare these models with the primate visual system, we used the Tensorflow library [Abadi et al., 2016] to augment standard CNNs with both local and long-range recurrence (Figure 1). Conduction from one area to another in visual cortex takes approximately 10 milliseconds (ms) [Mizuseki et al., 2009], with signal from photoreceptors reaching IT cortex at the top of the ventral stream by 70-100 ms. Neural dynamics indicating potential recurrent connections take place over the course of 100-200 ms [Issa et al., 2018]. A single feedforward volley of responses thus cannot be treated as if it were instantaneous relative to the timescale of recurrence and feedback. Hence, rather than treating each entire feedforward pass from input to output as one integral time step, as is normally done with RNNs [Spoerer et al., 2017], each time step in our models corresponds to a single feedforward layer processing its input and passing it to the next layer. This choice required an unrolling scheme different from that used in the standard Tensorflow RNN library, the code for which (and for all of our models) can be found at `https://github.com/neuroailab/tnn`.

Within each ConvRNN layer, feedback inputs from higher layers are resized to match the spatial dimensions of the feedforward input to that layer. Both types of input are processed by standard 2-D convolutions. If there is any local recurrence at that layer, the output is next passed to the recurrent cell as input. Feedforward and feedback inputs are combined within the recurrent cell (see Supplemental Methods). Finally, the output of the cell is passed through any additional nonlinearities,

such as max-pooling. The generic update rule for the discrete-time trajectory of such a network is thus $H_{t+1}^\ell = C_\ell \left( F_\ell \left( \bigoplus_{j \neq \ell} R_t^j \right), H_t^\ell \right)$, and $R_t^\ell = A_\ell(H_t^\ell)$, where $R_t^\ell$ is the output of layer $\ell$ at time $t$ and $H_t^\ell$ is the hidden state of the locally recurrent cell $C_\ell$ at time $t$. $A_\ell$ is the activation function and any pooling post-memory operations, and $\oplus$ denotes concatenation along the channel dimension with appropriate resizing to align spatial dimensions. The learned parameters of such a network consist of $F_\ell$, comprising any feedforward and feedback connections coming into layer $\ell = 1, \ldots, L$, and any of the learned parameters associated with the local recurrent cell $C_\ell$.

The simplest form of recurrence that we consider is the "Time Decay" model, which has a discrete-time trajectory given by $H_{t+1}^\ell = F_\ell \left( \bigoplus_{j \neq \ell} R_t^j \right) + \tau_\ell H_t^\ell$, where $\tau_\ell$ is the learned time constant at a given layer $\ell$. This model is intended to be a control for simplicity, where the time constants could model synaptic facilitation and depression in a cortical layer.

In this work, all forms of recurrence add parameters to the feedforward base model. Because this could improve task performance for reasons unrelated to recurrent computation, we trained two types of control model to compare to ConvRNNs: (1) Feedforward models with more convolution filters ("wider") or more layers ("deeper") to approximately match the number of parameters in a recurrent model; and (2) Replicas of each ConvRNN model unrolled for a minimal number of time steps, defined as the number that allows all model parameters to be used at least once. A minimally unrolled model has exactly the same number of parameters as its fully unrolled counterpart, so any increase in performance from unrolling longer can be attributed to recurrent computation. Fully and minimally unrolled ConvRNNs were trained with identical learning hyperparameters.

## 3 Results

**Novel RNN Cell Structures Improve Task Performance.** We first tested whether augmenting CNNs with standard RNN cells, vanilla RNNs and LSTMs, could improve performance on ImageNet object recognition (Figure 2a). We found that these cells added a small amount of accuracy when introduced into an AlexNet-like 6-layer feedforward backbone (Figure 2b). However, there were two problems with these recurrent architectures: first, these ConvRNNs did not perform substantially better than parameter-matched, minimally unrolled controls, suggesting that the observed performance gain was due to an increase in the number of unique parameters rather than recurrent computation. Second, making the feedforward model wider or deeper yielded an even larger performance gain than adding these standard RNN cells, but with fewer parameters. Adding other recurrent cell structures from the literature, including the UGRNN and IntersectionRNN [Collins et al., 2017], had similar effects to adding LSTMs (data not shown). This suggested that standard RNN cell structures, although well-suited for a range of temporal tasks, are less well-suited for inclusion within deep CNNs.

We speculated that this was because standard cells lack a combination of two key properties: (1) **Gating**, in which the value of a hidden state determines how much of the bottom-up input is passed through, retained, or discarded at the next time step; and (2) **Bypassing**, where a zero-initialized hidden state allows feedforward input to pass on to the next layer unaltered, as in the identity shortcuts of ResNet-class architectures (Figure 2a, top left). Importantly, both of these features are thought to address the problem of gradients vanishing as they are backpropagated to early time steps of a recurrent network or to early layers of a deep feedforward network, respectively. LSTMs employ gating, but no bypassing, as their inputs must pass through several nonlinearities to reach their output; whereas vanilla RNNs do bypass a zero-initialized hidden state, but do not gate their input (Figure 2a).

We thus implemented recurrent cell structures with both features to determine whether they function better than standard cells within CNNs. One example of such a structure is the "Reciprocal Gated Cell", which bypasses its zero-initialized hidden state and incorporates LSTM-like gating (Figure 2a, bottom right, see Supplemental Methods for the cell equations). Adding this cell to the 6-layer CNN improved performance substantially relative to both the feedforward baseline and minimally unrolled, parameter-matched control version of this model. Moreover, the Reciprocal Gated Cell used substantially fewer parameters than the standard cells to achieve greater accuracy (Figure 2b).

However, this advantage over LSTMs could have resulted accidentally from a better choice of training hyperparameters for the Reciprocal Gated Cells. It has been shown that different RNN structures can succeed or fail to perform a given task because of differences in trainability rather than differences in

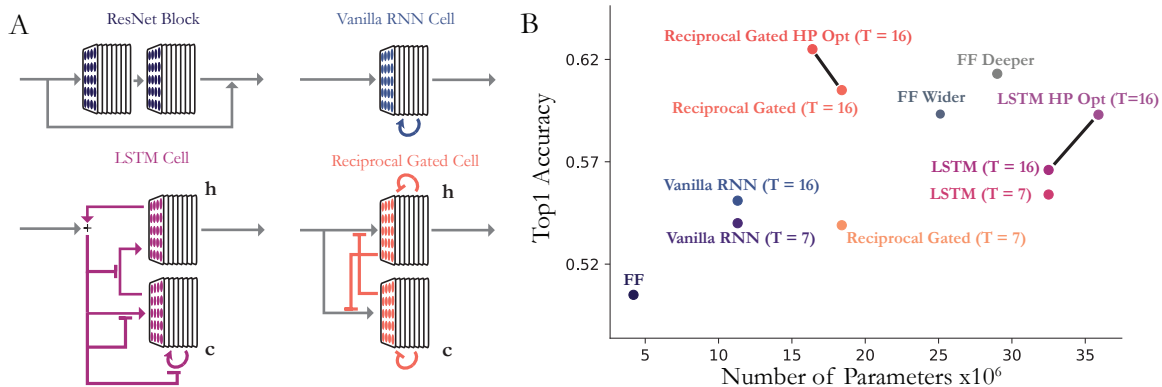

Figure 2: **Comparison of local recurrent cell architectures**. **(a) Architectural differences between ConvRNN cells.** Standard ResNet blocks and vanilla RNN cells have bypassing (see text). The LSTM cell has gating, denoted by "T"-junctions, but no bypassing. The Reciprocal Gated Cell has both. **(b) Performance of various ConvRNN and feedforward models as a function of number of parameters.** Colored points incorporate the respective RNN cell into the the 6-layer feedforward architecture ("FF"). "T" denotes number of unroll steps. Hyperparameter-optimized versions of the LSTM and Reciprocal Gated Cell ConvRNNs are connected to their non-optimized versions by black lines.

capacity [Collins et al., 2017]. To test whether the LSTM models underperformed for this reason, we searched over training hyperparameters and common structural variants of the LSTM to better adapt this local structure to deep convolutional networks, using hundreds of second generation Google Cloud Tensor Processing Units (TPUv2s). We searched over learning hyperparameters (e.g. gradient clip values, learning rate) as well as structural hyperparameters (e.g. gate convolution filter sizes, channel depth, whether or not to use peephole connections, etc.). While this did yield a better LSTM-based ConvRNN, it did not eclipse the performance of the smaller Reciprocal Gated Cell-based ConvRNN. Moreover, applying the same hyperparameter optimization procedure to the Reciprocal Gated Cell models equally increased that architecture class's performance and further reduced its parameter count (Figure 2b). Taken together, these results suggest that the choice of local recurrent structure strongly affects the accuracy of a ConvRNN on a challenging object recognition task. Although previous work has shown that standard RNN structures can improve performance on simpler object recognition tasks, these gains have not been shown to generalize to more challenging datasets such as ImageNet [Liao and Poggio, 2016, Zamir et al., 2017]. Differences between architectures may only appear on difficult tasks.

**Hyperparameter Optimization for Deeper Recurrent Architectures.** The success of Reciprocal Gated Cells on ImageNet prompted us to search for even better-adapted recurrent structures in the context of deeper feedforward backbones. In particular, we designed a novel search space over ConvRNN architectures that generalized that cell's use of gating and bypassing. To measure the quality of a large number of architectures, we installed RNN cell variants in a convolutional network based on the ResNet-18 model [He et al., 2016]. Deeper ResNet models, such as ResNet-34, reach higher performance on ImageNet and therefore provided a benchmark for how much recurrent computation could substitute for feedforward depth in this model class. Recurrent architectures varied in (a) how multiple inputs and hidden states were combined to produce the output of each cell; (b) which linear operations were applied to each input or hidden state; (c) which point-wise nonlinearities were applied to each hidden state; and (d) the values of additive and multiplicative biases (Figure 3a). The original Reciprocal Gated Cells fall within this larger hyperparameter space. We simultaneously searched over all possible long-range feedback connections, which use the output of a later layer as input to an earlier ConvRNN layer or cell hidden state. Finally, we varied learning hyperparameters of the network, such as learning rate and number of time steps to present the input image.

Since we did not want to assume how hyperparameters would interact, we jointly optimized architectural and learning hyperparameters using a Tree-structured Parzen Estimator (TPE) [Bergstra et al., 2011], a type of Bayesian algorithm which searches over continuous and categorical variable configurations (see Supplemental Methods for details). We trained hundreds of models simultaneously,

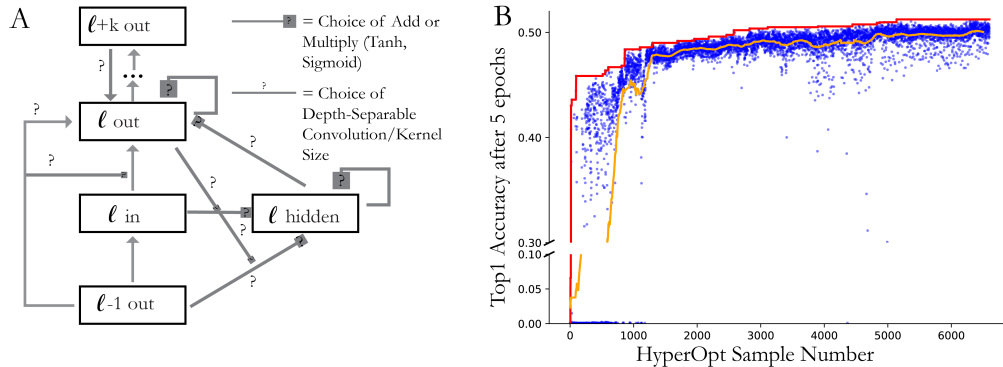

Figure 3: **ConvRNN hyperparametrization and search results. (a) Hyperparametrization of local recurrent cells.** Arrows indicate connections between cell input, hidden states, and output. Question marks denote optional connections, which may be conventional or depth-separable convolutions with a choice of kernel size. Feedforward connections between layers ($\ell - 1$ out, $\ell$ in, and $\ell$ out) are always present. Boxes with question marks indicate a choice of multiplicative gating with sigmoid or tanh nonlinearities, addition, or identity connections (as in ResNets). Finally, long-range feedback connections from $\ell + k$ out may enter at either the local cell input, hidden state, or output. **(b) ConvRNN search results.** Each blue dot represents a model, sampled from hyperparameter space, trained for 5 epochs. The orange line is the average performance of the last 50 models. The red line denotes the top performing model at that point in the search.

each for 5 epochs. To reduce the amount of search time, we trained these sample models on 128 px images, which yield nearly equal performance as larger images [Mishkin et al., 2016].

Most models in this architecture space failed. In the initial TPE phase of randomly sampling hyperparameters, more than 80% of models either failed to improve task performance above chance or had exploding gradients (Figure 3b, bottom left). However, over the course of the search, an increasing fraction of models reached higher performance. The median and peak sample performance increased over the next $\sim 7000$ model samples, with the best model reaching performance >15% higher than the top model from the initial phase of the search (Figure 3b). Like the contrast between LSTMs and Reciprocal Gated Cells, these dramatic improvements further support our hypothesis that some recurrent architectures are substantially better than others for object recognition at scale.

Sampling thousands of recurrent architectures allowed us to ask which structural features contribute most to task performance and which are relatively unimportant. Because TPE does not sample models according to a stationary parameter distribution, it is difficult to infer the causal roles of individual hyperparameters. Still, several clear patterns emerged in the sequence of best-performing architectures over the course of the search (Figure 3b, red line). These models used both bypassing and gating within their local RNN cells; depth-separable convolutions for updating hidden states, but conventional convolutions for connecting bottom-up input to output (Figure 4a); and several key long-range feedbacks, which were "discovered" early in the search, then subsequently used in the majority of models (Figure 4b). Overall, these recurrent structures act like feedforward ResNet blocks on their first time step but incorporate a wide array of operations on future time steps, including many of the same ones selected for in other work that optimized feedforward modules [Zoph et al., 2017]. Thus, this diversity of operations and pathways from input to output may reflect a natural solution for object recognition, whether implemented through recurrent or feedforward structures.

If the primate visual system uses recurrence in lieu of greater network depth to perform object recognition, then a shallower recurrent model with a suitable form of recurrence should achieve recognition accuracy equal to a deeper feedforward model [Liao and Poggio, 2016]. We therefore tested whether our search had identified such well-adapted recurrent architectures by fully training a representative ConvRNN, the model with the median five-epoch performance after 5000 TPE samples. This median model reached a final Top1 ImageNet accuracy nearly equal to a ResNet-34 model with nearly twice as many layers, even though the ConvRNN used only $\sim 75\%$ as many parameters (ResNet-34, 21.8 M parameters, 73.1% Validation Top1; Median ConvRNN, 15.5 M parameters, 72.9% Validation Top1). The fully unrolled model from the random phase of the search did not

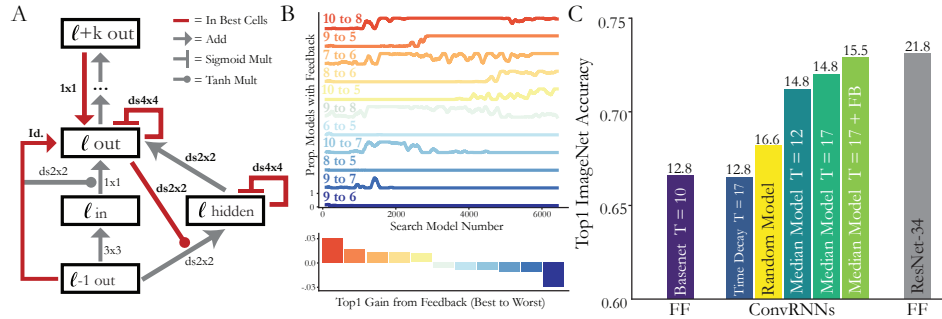

Figure 4: **Optimal local recurrent cell motif and global feedback connectivity. (a) RNN Cell structure from the top-performing search model.** Red lines indicate that this hyperparameter choice (connection and filter size) was chosen in each of the top unique models from the search (red line in Figure 3b). $K \times K$ denotes a convolution and $\mathrm{ds}K \times K$ denotes a depth-separable convolution with filter size $K \times K$. **(b) Long-range feedback connections from the search.** (Top) Each trace shows the proportion of models in a 100-sample window that have a particular feedback connection. (Bottom) Each bar indicates the difference between the median performance of models with a given feedback and the median performance of models without that feedback. Colors correspond to the same feedbacks as above. **(c) Performance of models fully trained on 224px-size ImageNet.** We compared the performance of an 18-layer feedforward base model (basenet) modeled after ResNet-18, a model with trainable time constants ("Time Decay") based on this basenet, the median model from the search with or without global feedback connectivity, and its minimally-unrolled control ($T = 12$). The "Random Model" was selected randomly from the initial, random phase of the model search. Parameter counts in millions are shown on top of each bar. ResNet models were trained as in [He et al., 2016] but with the same batch size of 64 to compare to ConvRNNs.

perform substantially better than the basenet, despite using more parameters – though the minimally unrolled control did (Figure 4c). We also considered a control model ("Time Decay"), described in Section 2, that produces temporal dynamics by learning time constants on the activations at each layer, rather than by learning connectivity between units. However, this model did not perform any better than the basenet, implying that ConvRNN performance is not a trivial result of outputting a dynamic time course of responses. Together these results demonstrate that the median model uses recurrent computations to perform object recognition and that particular motifs in its recurrent architecture, found through hyperparameter optimization, are required for its improved accuracy. Thus, given well-selected forms of local recurrence and long-range feedback, a ConvRNN can do the same challenging visual task as a deeper feedforward CNN.

**ConvRNNs Capture Neuronal Dynamics in the Primate Visual System.** ConvRNNs produce a dynamic time series of outputs given a static input. While these recurrent dynamics could be used for tasks involving time, here we optimized the ConvRNNs to perform the "static" task of object classification on ImageNet. It is possible that the primate visual system is optimized for such a task, because even static images reliably produce dynamic neural response trajectories [Issa et al., 2018]. Objects in some images become decodable from the neuronal population later than objects in other images, even though animals recognize both object sets equally well. Interestingly, late-decoding images are not well classified by feedforward CNNs, raising the possibility that they are encoded in animals through recurrent computations [Kar et al., 2018]. If this were the case, then recurrent networks trained to perform a difficult, static object recognition task might explain neural dynamics in the primate visual system, just as feedforward models explain time-averaged responses [Yamins et al., 2014, Khaligh-Razavi and Kriegeskorte, 2014]. We therefore asked whether a task-optimized ConvRNN could predict the firing dynamics of primate neurons recorded during encoding of visual stimuli. These data comprise multi-unit array recordings from the ventral stream of rhesus macaques during Rapid Serial Visual Presentation (RSVP, see Supplemental Methods [Majaj et al., 2015]).

We presented the fully trained median ConvRNN and its feedforward baseline with the same images shown to monkeys and collected the time series of features from each model layer. To decide which layer should be used to predict which neural responses, we fit linear models from each feedforward layer's features to the neural population and measured where explained variance on held-out images peaked (see Supplemental Methods). Units recorded from distinct arrays – placed in the successive

Figure 5: **Modeling primate ventral stream neural dynamics with ConvRNNs.** (a) The ConvRNN model used to fit neural dynamics has local recurrent cells at layers 4 through 10 and the long-range feedbacks denoted by the red arrows. (b) Consistent with the brain's ventral hierarchy, most units from V4 are best fit by layer 6 features; pIT by layer 7; and cIT/aIT by layers 8/9. (c) Fitting model features to neural dynamics approaches the noise ceiling of these responses. The y-axis indicates the median across units of the correlation between predictions and ground-truth responses on held-out images.

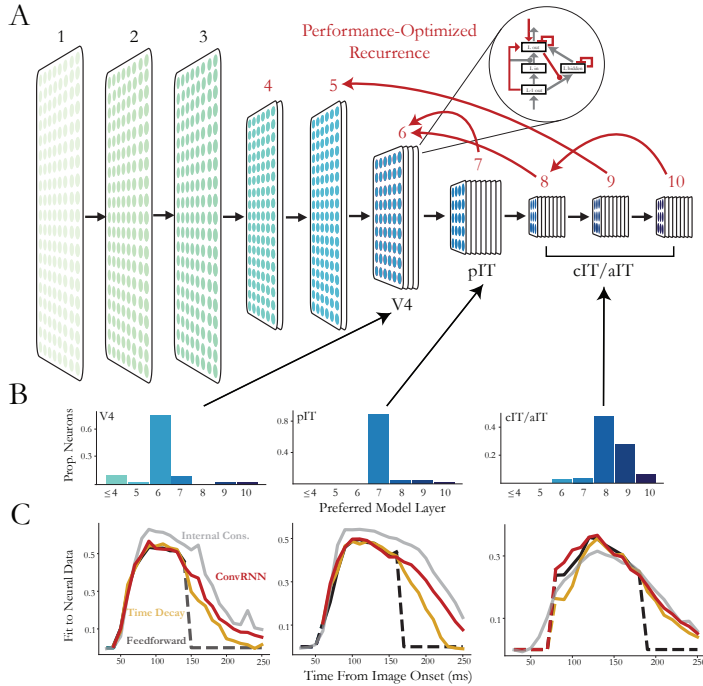

V4, posterior IT, and central/anterior IT cortical areas of the macaque – were fit best by the successive 6th, 7th, and 8/9th layers of the feedforward model, respectively (Figure 5a,b). This result refines the mapping of CNN layers to defined anatomical areas within the ventral visual pathway [Yamins et al., 2014]. Finally, we measured how well the ConvRNN features from these layers predicted the dynamics of each unit. Critically, we used the same linear mapping across all times so that the task-optimized dynamics of the recurrent network had to account for the neural dynamics; each recorded unit trajectory was fit as a linear combination of model unit trajectories.

On held-out images, the ConvRNN feature dynamics fit the neural response trajectories as well as or better than the feedforward baseline features for almost every 10 ms time bin (Figure 5c). Moreover, because the feedforward model has the same square wave dynamics as its 100ms visual input, it cannot predict neural responses after image offset plus a fixed delay, corresponding to the number of layers (Figure 5c, dashed black lines). In contrast, the activations of ConvRNN units have persistent dynamics, yielding strong predictions of the entire neural response. This improvement is especially clear when comparing the predicted temporal responses of the feedforward and ConvRNN models on single images (Figure S1a): ConvRNN predictions have dynamic trajectories that resemble smoothed versions of the ground truth responses, whereas feedforward predictions necessarily have the same time course for every image. Crucially, this results from the nonlinear dynamics learned on ImageNet, as both models are fit to neural data with the same form of temporally-shared linear model and the same number of parameters. Across held-out images, the ConvRNN outperforms the feedforward model in its prediction of single-image temporal responses (Figure S1b).

Since the initial phase of neural dynamics was well-fit by feedforward models, we asked whether the later phase could be fit by a much simpler model than the ConvRNN, namely the Time Decay model with ImageNet-trained time constants at convolutional layers. In this model, unit activations have exponential rather than immediate falloff once feedforward drive ceases. These dynamics could arise from single-neuron biophysics (e.g. synaptic depression) rather than interneuronal connections. If the Time Decay model were to explain neural data as well as ConvRNNs, it would imply that interneuronal recurrent connections are not needed to account for the observed dynamics; however, this model did not fit the late phase dynamics of intermediate areas (V4 and pIT) or the early phase of cIT/aIT responses as well as the ConvRNN (Figure 5c). Thus, the more complex recurrence found in ConvRNNs is needed both to improve object recognition performance and to account for neural dynamics in the ventral stream, even when animals are only required to fixate on visual stimuli.

# 4   Discussion

Here, we show that careful introduction of recurrence into CNNs can improve performance on a challenging visual task. Even hyperparameter optimization of traditional recurrent motifs yielded ImageNet performance inferior to hand-designed motifs. Our findings raise the possibility that different locally recurrent structures within the brain may be adapted for different behaviors. In this case, the combination of locally recurrent gating, bypassing, and depth-separability was required to improve ImageNet performance. Although we did not explicitly design these structures to resemble cortical microcircuits, the presence of reciprocal, cell type-specific connections between discrete neuronal populations is consistent with known anatomy [Costa et al., 2017]. With the proper local recurrence in place, specific patterns of long-range feedback connections also increased performance. This may indicate that long-range connections in visual cortex are important for object recognition.

ConvRNNs optimized for the ImageNet task strongly predicted neural dynamics in macaque V4 and IT. Specifically, the temporal trajectories of neuronal firing rates were well-fit by linear regression from task-optimized ConvRNN features. Thus we extend a core result from the feedforward to the recurrent regime: given an appropriate architecture class, task-driven convolutional recurrent neural networks provide the best normative models of encoding dynamics in the primate visual system.

The results of fitting ConvRNN features to neural dynamics are somewhat surprising in two respects. First, ConvRNN predictions are significantly stronger than feedforward CNN predictions only at time points when the feedforward CNNs have no visual response (Figure 5c). This implies that early dynamics in V4 and IT are, on average across stimuli, dominated by feedforward processing. However, many neural responses to single images do not have the square-wave trajectory output by a CNN (Figure S1a); for this reason, single-image ConvRNN trajectories are much more highly correlated with ground truth trajectories across time (Figure S1). Although some firing rate dynamics may result from neuronal biophysics (e.g. synaptic depression and facilitation) rather than recurrent connections between neurons, models with task-optimized neuronal time constants at each convolutional layer did not predict neural dynamics as well as ConvRNNs (Figure 5c). This simple form of "recurrence" also failed to improve ImageNet performance. Thus, non-trivial recurrent connectivity is likely needed to explain the visually-evoked responses observed here. Since model predictions do not reach the internal consistency of the responses, further elaborations of network architecture or the addition of new tasks may yield further improvements in fitting. Alternatively, the internal consistency computed here may incorporate task-irrelevant neural activity, such as from animal-specific anatomy or circuits not functionally engaged during RSVP. These models will therefore need to be tested on neural dynamics of animals performing active visual tasks.

Second, it has often been suggested that recurrence is important for processing images with particular semantic properties, such as clutter or occlusion. Here, in contrast, we have learned recurrent structures that improve performance on the same general categorization task performed by feedforward networks, without *a priori* mandating a "role" for recurrence that is qualitatively distinct from that of any other model component. A specialized role for recurrence might be identified by analyzing the learned motifs *ex post facto*. However, recent electrophysiology experiments suggest this might not be the most effective approach. Kar et al. [2018] have identified a set of "challenge images" on which feedforward CNNs fail to solve object recognition tasks, but which are easily solved by humans and non-human primates. Moreover, objects in challenge images are decodable from IT cortex population responses significantly later in the neuronal timecourse than control images, suggesting that recurrent circuits are meaningfully engaged when processing the challenge images. Interestingly, however, the challenge images appear *not* to have any clear defining properties or language-level description, such as containing occlusion, clutter, or unusual object poses. This observation is consistent with our results, in which the non-specific "role" of recurrence in ConvRNNs is simply to help recognize objects across the wide variety of conditions present in natural scenes. Since biological neural circuits have been honed by natural selection on animal behavior, not on human intelligibility, the role of an individual circuit component may in general have no simple description.

Optimizing recurrent augmentations for more sophisticated feedforward CNNs, such as NAS-Net [Zoph et al., 2017], could lead to improvements to these state-of-the-art baselines. Further experiments will explore whether different tasks, such as those that do not rely on difficult-to-obtain category labels, can substitute for supervised object recognition in matching ConvRNNs to neural responses.

# 5 Acknowledgments

This project has received funding from the James S. McDonnell foundation (grant #220020469, D.L.K.Y.), the Simons Foundation (SCGB grants 543061 (D.L.K.Y) and 325500/542965 (J.J.D)), the US National Science Foundation (grant IIS-RI1703161, D.L.K.Y.), the European Union's Horizon 2020 research and innovation programme (agreement #705498, J.K.), the US National Eye Institute (grant R01-EY014970, J.J.D.), and the US Office of Naval Research (MURI-114407, J.J.D). We thank Google (TPUv2 team) and the NVIDIA corporation for generous donation of hardware resources.

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
