[Supplementary Material · neurips_2018_supplementary.pdf]

# 6 Supplemental Methods

## 6.1 CNN architectures

For all of our CNN architectures, we use Elu nonlinearities. The 6-layer feedforward CNN, referenced in Figure 2b as "FF", had the following architecture:

| Layer | Kernel Size | Channels | Stride | Max Pooling |
|-------|-------------|----------|--------|-------------|
| 1 | $7 \times 7$ | 64 | 2 | $2 \times 2$ |
| 2 | $3 \times 3$ | 128 | 1 | $2 \times 2$ |
| 3 | $3 \times 3$ | 256 | 1 | $2 \times 2$ |
| 4 | $3 \times 3$ | 256 | 1 | $2 \times 2$ |
| 5 | $3 \times 3$ | 512 | 1 | $2 \times 2$ |
| 6 | $2 \times 2$ | 1000 | 1 | No |

The variant of the above 6-layer feedforward CNN, referenced in Figure 2b as "FF wider" is given below:

| Layer | Kernel Size | Channels | Stride | Max Pooling |
|-------|-------------|----------|--------|-------------|
| 1 | $7 \times 7$ | 128 | 2 | $2 \times 2$ |
| 2 | $3 \times 3$ | 512 | 1 | $2 \times 2$ |
| 3 | $3 \times 3$ | 512 | 1 | $2 \times 2$ |
| 4 | $3 \times 3$ | 512 | 1 | $2 \times 2$ |
| 5 | $3 \times 3$ | 1024 | 1 | $2 \times 2$ |
| 6 | $2 \times 2$ | 1000 | 1 | None |

The "FF deeper" model referenced in Figure 2b is given below:

| Layer | Kernel Size | Depth | Stride | Max Pooling |
|-------|-------------|-------|--------|-------------|
| 1 | $7 \times 7$ | 64 | 2 | $2 \times 2$ |
| 2 | $3 \times 3$ | 64 | 1 | None |
| 3 | $3 \times 3$ | 64 | 1 | None |
| 4 | $3 \times 3$ | 128 | 1 | $2 \times 2$ |
| 5 | $3 \times 3$ | 128 | 1 | None |
| 6 | $3 \times 3$ | 256 | 1 | $2 \times 2$ |
| 7 | $3 \times 3$ | 256 | 1 | $2 \times 2$ |
| 8 | $3 \times 3$ | 512 | 1 | None |
| 9 | $3 \times 3$ | 512 | 1 | None |
| 10 | $3 \times 3$ | 512 | 1 | $2 \times 2$ |
| 11 | None (Avg. Pool FC) | 1000 | None | None |

The 18 layer CNN modeled after ResNet-18 (using MaxPooling rather than stride-2 convolutions to perform downsampling) is given below:

| Block | Kernel Size | Depth | Stride | Max Pooling | Repeat |
|-------|-------------|-------|--------|-------------|--------|
| 1 | $7 \times 7$ | 64 | 2 | $2 \times 2$ | $\times 1$ |
| 2 | $3 \times 3$ | 64 | 1 | None | $\times 2$ |
| 3 | $3 \times 3$ | 64 | 1 | None | $\times 2$ |
| 4 | $3 \times 3$ | 128 | 1 | $2 \times 2$ | $\times 2$ |
| 5 | $3 \times 3$ | 128 | 1 | None | $\times 2$ |
| 6 | $3 \times 3$ | 256 | 1 | $2 \times 2$ | $\times 2$ |
| 7 | $3 \times 3$ | 256 | 1 | $2 \times 2$ | $\times 2$ |
| 8 | $3 \times 3$ | 512 | 1 | None | $\times 2$ |
| 9 | $3 \times 3$ | 512 | 1 | None | $\times 2$ |
| 10 | $3 \times 3$ | 512 | 1 | $2 \times 2$ | $\times 2$ |
| 11 | None (Avg. Pool FC) | 1000 | None | None | $\times 1$ |

We additionally used an auxiliary classifier located at 70% of the way up the network, similar to [Zoph et al., 2017].

## 6.2 Reciprocal Gated Cell Equation

We will provide the update equations for the "Reciprocal Gated Cell", diagrammed in Figure 2a (bottom right). Let $\circ$ denote Hadamard (elementwise) product, let $*$ denote convolution, and let $x_t^\ell$ denote the input to the cell at layer $\ell$. Then the update equations are as follows:

The update equation for the output of the cell, $h_t^\ell$, is given by a gating of both the input and memory $c_t^\ell$.

$$
\begin{aligned}
a_{t+1}^\ell &= (1 - \sigma(W_{ch}^\ell * c_t^\ell)) \circ x_t^\ell + (1 - \sigma(W_{hh}^\ell * h_t^\ell)) \circ h_t^\ell \\
h_t^\ell &= f\left(a_t^\ell\right).
\end{aligned}
\tag{1}
$$

The update equation for the memory $c_t^\ell$ is given by a gating of the input and the output of the cell $h_t^\ell$.

$$
\begin{aligned}
\tilde{c}_{t+1}^\ell &= (1 - \sigma(W_{hc}^\ell * h_t^\ell)) \circ x_t^\ell + (1 - \sigma(W_{cc}^\ell * c_t^\ell)) \circ c_t^\ell \\
c_t^\ell &= f(\tilde{c}_t^\ell).
\end{aligned}
\tag{2}
$$

## 6.3 RNN Cell Search Parametrization

Inspired by the Reciprocal Gated Cell equations above, there are a variety of possibilities for how $h_{t-1}^\ell, x_t^\ell, c_t^\ell$, and $h_t^\ell$ can be connected to one another. Figure 3a schematizes these possibilities.

Mathematically, Figure 3a can be formalized in terms of the following update equations. First, we define our input sets and building block functions:

$$
\begin{aligned}
minin &= \{h_{t-1}^{\ell-1}, x_t^\ell, c_{t-1}^\ell, h_{t-1}^\ell\} \\
minin_a &= minin \cup \{c_t^\ell\} \\
minin_b &= minin \cup \{h_t^\ell\} \\
S_a &\subseteq minin_a \\
S_b &\subseteq minin_b \\
\text{Affine}(x) &\in \{+, 1 \times 1 \text{ conv}, K \times K \text{ conv}, K \times K \text{ depth-separable conv}\} \\
K &\in \{3, \dots, 7\}
\end{aligned}
$$

With those in hand, we have the following update equations:

$$
\begin{aligned}
\tau_a &= v_1^\tau + v_2^\tau \sigma(\text{Affine}(S_a)) \\
\tau_b &= v_1^\tau + v_2^\tau \sigma(\text{Affine}(S_b)) \\
gate_a &= v_1^g + v_2^g \sigma(\text{Affine}(S_a)) \\
gate_b &= v_1^g + v_2^g \sigma(\text{Affine}(S_b)) \\
a_t^\ell &= \{gate_a\} \cdot in_t^\ell + \{\tau_a\} \cdot h_{t-1}^\ell \\
h_t^\ell &= f(a_t^\ell) \\
b_t^\ell &= \{gate_b\} \cdot in_t^\ell + \{\tau_b\} \cdot c_{t-1}^\ell \\
c_t^\ell &= f(b_t^\ell) \\
f &\in \{\text{elu}, \tanh, \sigma\}.
\end{aligned}
$$

For clarity, the following matrix summarizes the connectivity possibilities (with ? denoting the possibility of a connection):

$$
\begin{array}{c}
\begin{array}{cccccc}
\quad h_{t-1}^{\ell-1} & x_t^\ell & c_{t-1}^\ell & c_t^\ell & h_{t-1}^\ell & h_t^\ell
\end{array}\\
\begin{array}{c}
h_{t-1}^{\ell-1}\\
x_t^\ell\\
c_{t-1}^\ell\\
c_t^\ell\\
h_{t-1}^\ell\\
h_t^\ell
\end{array}
\left(
\begin{array}{cccccc}
0 & 1 & 0 & ? & 0 & ?\\
0 & 0 & 0 & ? & 0 & ?\\
0 & 0 & 0 & ? & 0 & ?\\
0 & 0 & 0 & 0 & 0 & ?\\
0 & 0 & 0 & ? & 0 & ?\\
0 & 0 & 0 & ? & 0 & 0
\end{array}
\right)
\end{array}
$$

## 6.4 Training ConvRNN Models

ConvRNN models were trained by stochastic gradient descent on 128 px ImageNet using the same learning rate schedule and data augmentation methods as the original ResNet models [He et al., 2016] with the following modifications: we used a batch size of 128, an initial learning rate of 0.01, and Nesterov momentum of 0.9. Standard ImageNet cross-entropy loss was computed from the last time step of a ConvRNN. Gradients with absolute value greater than 0.7 were clipped.

## 6.5 Hyperparameter Optimization Methods

Models were trained synchronously 100 models at a time using the HyperOpt package [Bergstra et al., 2015], for 5 epochs each. Each model was trained on its own Tensor Processing Unit (TPUv2), and around 6000 models were sampled in total over the course of the search. During the search, ConvRNN models were trained by stochastic gradient descent on 128 px ImageNet for efficiency. The top performing ConvRNN models were then fully trained out on 224 px ImageNet with a batch size of 64 (which was maximum that we could fit into memory), and the ResNet models were also trained using this same batch size for fair comparison.

We employed a form of Bayesian optimization, a Tree-structured Parzen Estimator (TPE), to search the space of continuous and categorical hyperparameters [Bergstra et al., 2011]. This algorithm constructs a generative model of $P[score \mid configuration]$ by updating a prior from a maintained history $H$ of hyperparameter configuration-loss pairs. The fitness function that is optimized over models is the expected improvement, where a given configuration $c$ is meant to optimize $EI(c) = \int_{x<t} P[x \mid c, H]$. This choice of Bayesian optimization algorithm models $P[c \mid x]$ via a Gaussian mixture, and restricts us to tree-structured configuration spaces.

## 6.6 Fitting Model Features to Neural Data

We fit trained model features to multi-unit array responses from [Majaj et al., 2015]. Briefly, we fit to 256 recorded sites from two monkeys. These came from three multi-unit arrays per monkey: one implanted in V4, one in posterior IT, and one in central and anterior IT. Each image was presented approximately 50 times, using rapid visual stimulus presentation (RSVP). Each stimulus was presented for 100 ms, followed by a mean gray background interleaved between images. Each trial lasted 250 ms. The image set consisted of 5120 images based on 64 object categories. Each image consisted of a 2D projection of a 3D model added to a random background. The pose, size, and $x$- and $y$-position of the object was varied across the image set, whereby 2 levels of variation were used (corresponding to medium and high variation from [Majaj et al., 2015]). Multi-unit responses to these images were binned in 10ms windows, averaged across trials of the same image, and normalized to the average response to a blank image. This produced a set of (5120 images x 256 units x 25 time bins) responses, which were the targets for our model features to predict. To estimate a noise ceiling for each mean response at each time bin, we computed the Spearman split-half reliability across trials.

The 5120 images were split 75-25 within each object category into a training set and a held-out testing set. All images were presented to the models for 10 time steps (corresponding to 100 ms), followed by a mean gray stimulus for the remaining 15 time steps, to match the image presentation to the monkeys.

Model features from each image (i.e. the activations of units in a given model layer) were linearly fit to the neural responses by stochastic gradient descent with a standard L2 loss. Each of the 256

Figure S1: **Comparison of feedforward models and ConvRNNs to single images.** (a) Examples of a single pIT unit's response to three different images from the held-out validation set. Dashed lines are the ground truth response, solid lines are the model prediction. The feedforward model is forced to have the dynamics of a square wave. (b) Correlations of ground truth with predicted responses across time bins (50 to 250 ms). The temporal correlation is computed for each image and for each unit. Plotted: the median across units from each array of the median temporal correlation across single held-out validation images.

units was fit independently. We stipulated that units from each multi-unit array must be fit by features from a single model layer. To determine which one, we fit the features from the feedforward model to each time bin of a given unit's response, averaged across time bins, and counted how many units had minimal loss for a given model layer. This yielded a mapping from the V4 array to model layer 6, pIT to layer 7, and cIT/aIT to layer 8. Finally, model features were fit to the entire set of 25 time bins for each unit using a shared linear model: that is, a single set of regression coefficients was used for all time bins. The loss for this fitting was the average L2 loss across training images and 25 time bins for each unit.