[Reviews · NeurIPS 2018]

Reviewer 1



Post author feedback: I am very impressed by the fits at the bottom of the response. There was some discussion amongst the reviewers concerning the relationship between this and what is known about the actual circuits (e.g., inputs arrive to layers 4 and 5, then from layer 4 signals go to layers 2/3, etc.). It would be useful for the authors to relate this to those facts. Also, we discussed whether your model actually fits the data about the quantity of feedback vs. feedforward connections (as much or more feedback as feedforward). It would be useful to inform the reader as to whether your model accounts for this as well. Summary: This paper takes inspiration from the primate visual system, which, unlike standard convolutional networks, includes feedback and lateral connections. Indeed, while CNNs are the best model we have of the temporal lobe, they cannot account for the temporal processing in animal visual systems. There is evidence (referenced here) that for difficult-to-recognize images, the neural activity doesn’t settle down right away, suggesting a major role for feedback. There are several possible roles for this feedback, including dealing with occlusion, cleaning up noisy inputs, etc. The first thing they do to ameliorate this situation is try to include LSTM and vanilla recurrence in a six-layer Alex-net-like model. These do not improve the accuracy. However, a new type of recurrent cell, a Reciprocal Gated Cell, that includes both bypass connections and gating connections, was able to improve performance. Encouraged by this success, they initiated a huge search of potential recurrent cell architectures over ResNet-18 that included variations in connection architecture, activation function types, and arithmetic operations. They found that the median optimal architecture, when trained on imagenet, achieved the same top-1 performance as ResNet-36, with half as many layers and 75% fewer parameters. They compared these architectures to strong baselines, including feed forward networks with the same amount of computations, but no recurrence, and found that the recurrent architecture was still superior. Finally, the coup-de-grace was applying this architecture to accounting for the time course of neural responses (at 10ms intervals) in monkey visual cortex. They found that they were close to the noise limit in fitting this data, and did so much better than a feed-forward network. Quality: Strengths: This is excellent work, and shows the power of having 100s of TPUs at your disposal, in the right hands. The role of recurrent processing in the cortex has long been debated, but this kind of work not only extends convolutional neural networks in a more biologically-plausible direction, but promises to reveal the role of this processing. It also shows how we can account for the time course of neural responses with such a model. Weaknesses: I could not detect any. Clarity: The paper is very clear, and this reviewer especially appreciated page 2, which pretty much summarized the entire contribution. However, that said, the notation used in the paper, especially on line 113, is terrible. The same notation (uppercase italic letters) is used for weights, activations, and activation functions. I strongly suggest making this clearer. Also, in the supplementary material, the more detailed equations are good, except that there are still undefined quantities and a few puzzles. For example, it is clear how the gating works in equation 1 amnd 2, but in the equations following line 429, it is unclear why the gates should actually be between 0 and 1, unless they are not supposed to be restricted in such a way, and nome of the v variables are defined. Are these the learnable parameters? Originality: Clearly, this has not been done before. Significance: This is highly significant. The issue of adding recurrence to convolutional networks has been around for a while, and this is the first paper I’ve seen that does it very effectively. This also has implications for understanding how the brain works. The replication question below hinges a great deal on what you mean by a replication. Their code will be available, but if someone wanted to repeat their search process, it would be difficult without access to 100’s of second generation TPUs. I can’t help but wonder how many households could be powered for a year by the amount of energy used in this search.

Reviewer 2



Here is an update following the authors' response. After reading the authors' response, I strongly feel that this paper should be rejected. My main argument is that this paper is more engineering (indeed a tour de force) than science. My point is that there is a large body of work that has looked at the computational role of recurrent connections (some of it cited, some of it mentioned by the reviewers) and this work does not shed any light on the computational role of recurrent connections. If we look at the actual contributions of the paper in terms of engineering, I do not think the reported improvements are that impressive. We are literally talking about 1% compared to a ResNet-34 baseline which is far from the state of the art. Furthermore -- yes the baseline has 75% more parameters but we are not talking orders of magnitude here. Would the proposed circuit help push the current state of the art? Well we dont know. One potential key limitation brushed under the carpet is that the proposed module come with a very significant increase in run-time complexity / FLOS (at least one order of magnitude more than the baselines) potentially limiting the applicability of the circuits for state of the art very deep architectures. Regarding the neuroscience side of the contribution. The point that I made in my review is that the neural data are probably far from ideal to test a recurrent model because of the passive viewing and rapid presentation paradigm. It could well be that the dynamics beyond the initial onset is actually trivial and the noise ceiling not all that hard to reach. My review had hinted at several simple feedforward models that need to be considered for baselines (e.g, synaptic depression) to show that the explained variance of their model is not trivially accounted by a time-decay model. The authors claim that they have run a time-decaying model as controls and that they "did not do so nearly as well as task-optimized ConvRNNs". This is up to the referees to decide the validity of this statement and I am unclear why the authors did not submit these results in their rebuttal. Also to be clear, the controls that I mentioned are more than a simple linear decay on the model output but would involve a simple decay of the weights that would translate in a non-linear decay on the neural output. Overall, the proposed model shows small improvements over non-SOA approaches with a very significant extra cost in FLOPS. In addition, the authors failed to demonstrate that their neural data exhibit any non-trivial dynamics and that the observed improvement in fitting these data could not be obtained with much simpler baselines. Overall, I see this paper as a clear reject and I am willing to lower my score and fight for rating should the need arise. *** This study builds on previous work by Dicarlo's group that has shown that i) it is possible to optimize hyperparameters of conv nets to yield significant gains in accuracy and that ii) in turn, this yields computational models that fit better to ventral stream neural data. Here the idea is to go one step further and using a similar optimization approach to extend the models to incorporate recurrent processes. Unfortunately, the study falls short on several fronts. From the network optimization perspective, the benefits of the approach are not quite as significant as those typically obtained with FF / conv nets. Furthermore, there is no clear computational-level explanation that can be gained because the dataset used is uncontrolled (imagenet) and it is not clear how recurrent processes contribute to improvements in accuracy (is it reducing noise? dealing with occlusions? clutter?). Recurrent connections help improve object recognition accuracy but then there is always a slightly deeper network (the number of parameters is of the same order of magnitude) that can perform object recognition with accuracy on par with these recurrent networks. In contrast, the previous cited work used synthetic stimuli and was able to show improvements with recurrent processing under occlusion etc. Here it is not clear what improvement truly is achieved and when. The study seems to be adding more noise than it is helping the field... Another contribution of the work is that the resulting recurrent networks can then be used to fit -- not just temporal averages of neural signals as done in earlier work by Yamins et al -- but the entire neural dynamics. However, it is not clear from the paper how truly significant this is. For one, the monkey is passively fixating and stimulus presentations are fast -- hence the dynamics could well be relatively trivial'' (i.e., initial transient followed by a gradual decay in firing). At the very least, the authors should have provided a few controls on the FF nets. For instance, adaptation and/or synaptic depression on the weights of the FF nets might be sufficient to explain the variance in the neural dynamics and hence recurrent processes are not really needed.

Reviewer 3



This paper introduces and evaluates neuroscience-inspired refinements of classic deep feedforward convolutional neural networks for object classification. Specifically, the authors experiment with new types of processing units which implement mechanisms of recurrence and top-down feedback within the network -- processes thought to be important functional mechanisms in biological vision. The paper is well-conceived and well-argued, and the model development and evaluation is extremely comprehensive and technically accomplished. Importantly, the authors demonstrate how their recurrent models achieve comparable performance to much deeper feedforward networks, while at the same time providing a better fit to neurophysiological data. Moreover, this paper is also an excellent demonstration of how processing principles inspired by neurocognitive theory can be investigated using machine learning and neural network modelling, leading to sophisticated, quantitative evaluation of hypotheses about the functional organisation of the brain (following, e.g., Cox, Seidenberg, & Rogers, 2015; Kriegeskorte, 2015). Detailed comments A clear strength of this paper is its comprehensive modelling rigor. For example, I particularly liked the use of carefully matched control networks (lines 119-128). Another example is the attempt to control for the trainability of different architectures through comprehensive hyperparameter tuning (lines 159-171). It is also great that the code for the models is being publically released. Reciprocal Gated Cells (and their parameterised generalization), introduced in this paper, would seem to have the potential to make an important fundamental contribution to computer vision engineering (in addition to yielding more neurocognitively plausible models). I think the authors could expand a little on the design choices leading to the Reciprocal Gated Cell. For example, couldn’t the standard LSTM cell be adapted to include bypassing? Is there an intuitive reason why the mutual, symmetric, gating by the h and c layers of each others’ inputs can be regarded, a priori, as a desirable property? My main critical comment is with the ethological relevance of an object labelling task (this is an issue for other similar work as well, not just this paper). As the authors note, primate vision has the goal of transforming the visual input into representations that “readily guide behaviour”. Is object labelling really the most suitable machine learning task for approximating successful behaviour? Why should accurate labelling of objects be the objective function of a model of biological visual systems? This seems particularly problematic for non-human primates, who do not use labels for objects at all, but nevertheless activate adaptation-optimised, behaviour-relevant high-level representations of objects. When a monkey sees either a lion or a leopard, presumably they activate a high-level representation we might gloss as “it’s dangerous; it’s fast; I need to get away right now”. This would be very different to the representation activated for, say, a banana. However, a labelling task will lead the network to learn representations that are best for visually discriminating between all objects (e.g. LION, LEOPARD, BANANA), not representations that reflect meaningful, behaviour-relevant similarities and differences between objects (representations in which LION and LEOPARD are presumably more similar to each other than to BANANA). It seems to me that training neural networks to decode categorical similarity or property knowledge rather than object labels (Devereux, Clarke, & Tyler, 2018; Jozwik, Kriegeskorte, Storrs, & Mur, 2017; Rabovsky & McRae, 2014; Rogers et al., 2004) may reflect more suitable objectives for networks intended to model meaningful object processing. Classic connectionist literature on concept learning (e.g. McClelland and others) may also be relevant here. However, I consider this criticism to be relatively minor, given the other significant contributions of the paper. Given the stated motivation to model biological visual systems, I would have liked to have seen the models evaluated against a wider range of neuroscience data, including fMRI (perhaps using Representational Similarity Analysis (Kriegeskorte, Goebel, & Bandettini, 2006), following, for example, the analyses of (Khaligh-Razavi & Kriegeskorte, 2014)). Comparison to EEG, ECoG, or MEG data would also have been interesting (e.g. Cichy, Khosla, Pantazis, Torralba, & Oliva, 2016), given the modelling focus on plausible dynamics (lines 101-105). Is it particularly surprising that vanilla RNNs and LSTMs do not increase performance on object labelling, since these architectures are designed for sequence modelling (which is not relevant to the object labelling task)? I would like to see a slightly more detailed discussion and rationale for why the authors considered these worth testing in the first place (in terms of how they might relate to known functional properties of the primate visual system). Perhaps some of the ideas in lines 143-158 belong earlier, in the introduction and methods sections. Final comments From a computational cognitive neuroscience perspective, it is very exciting that the recurrent models presented in this paper do as well as neuro-cognitively implausible ultra-deep feedforward models. This paper is a rarity in that it simultaneously makes fundamental contributions to both computer vision engineering and high-level computational neuroscience, and I therefore heartily recommend its acceptance. References Cichy, R. M., Khosla, A., Pantazis, D., Torralba, A., & Oliva, A. (2016). Comparison of deep neural networks to spatio-temporal cortical dynamics of human visual object recognition reveals hierarchical correspondence. Scientific Reports, 6, 27755. https://doi.org/10.1038/srep27755 Cox, C. R., Seidenberg, M. S., & Rogers, T. T. (2015). Connecting functional brain imaging and Parallel Distributed Processing. Language, Cognition and Neuroscience, 30(4), 380–394. https://doi.org/10.1080/23273798.2014.994010 Devereux, B. J., Clarke, A., & Tyler, L. K. (2018). Integrated deep visual and semantic attractor neural networks predict fMRI pattern-information along the ventral object processing pathway. Scientific Reports, 8(1), 10636. https://doi.org/10.1038/s41598-018-28865-1 Jozwik, K. M., Kriegeskorte, N., Storrs, K. R., & Mur, M. (2017). Deep Convolutional Neural Networks Outperform Feature-Based But Not Categorical Models in Explaining Object Similarity Judgments. Frontiers in Psychology, 8. https://doi.org/10.3389/fpsyg.2017.01726 Khaligh-Razavi, S.-M., & Kriegeskorte, N. (2014). Deep Supervised, but Not Unsupervised, Models May Explain IT Cortical Representation. PLoS Comput Biol, 10(11), e1003915. https://doi.org/10.1371/journal.pcbi.1003915 Kriegeskorte, N. (2015). Deep Neural Networks: A New Framework for Modeling Biological Vision and Brain Information Processing. Annual Review of Vision Science, 1(1), 417–446. https://doi.org/10.1146/annurev-vision-082114-035447 Kriegeskorte, N., Goebel, R., & Bandettini, P. (2006). Information-based functional brain mapping. Proceedings of the National Academy of Sciences of the United States of America, 103(10), 3863–3868. https://doi.org/10.1073/pnas.0600244103 Rabovsky, M., & McRae, K. (2014). Simulating the N400 ERP component as semantic network error: Insights from a feature-based connectionist attractor model of word meaning. Cognition, 132(1), 68–89. https://doi.org/10.1016/j.cognition.2014.03.010 Rogers, T. T., Lambon Ralph, M. A., Garrard, P., Bozeat, S., McClelland, J. L., Hodges, J. R., & Patterson, K. (2004). Structure and Deterioration of Semantic Memory: A Neuropsychological and Computational Investigation. Psychological Review, 111(1), 205–235. https://doi.org/10.1037/0033-295X.111.1.205

Reviewer 4



Taking into account discussion after author's response, I agree that questions about discussion and relationship to prior work are relatively minor. The most critical issue for this reviewer was that local recurrence was not modeled and across layers useful recurrent patterns were not summarized in a way that can inform the development of either experiments or theoretical (analytical) studies. This submission addresses the question of how recurrent interactions can improve object classification performance of deep convolutional networks. Authors find that certain long-range recurrent interactions that skip layers improve performance on an object classification task. The weaknesses of this work is that it strongly overstates its claims and does not provide error-bars on performance from different initialization states. Another weakness is that the type of recurrence that is reported to be beneficial -- recurrent interactions across layers -- does not resemble local recurrence for which cortical circuits are known for. Finally, the introduction and discussion of the results are based on a large number of unpublished studies from the arxiv and ignores results from peer-reviewed literature on simpler and more closely related to neural circutry models than convolutional deep networks. Examples of these papers by Nishimoto and Gallant J Neurosci 2011 who systematically tested the contribution of different types of nonlinearities and ultimately determined a non-convolutional model to be the optimal one within the tested family of models. Similarly, a non-convolutional models was selected by Liu et al PNAS 2016 as the one achieving best predictive power in V2.